An analysis of the 24-hour on-call experience and treatment decision of a dental resident, a retrospective study

Zheng Jiaoer
Xu Ji
http://orcid.org/0000-0002-4429-6275 Zhang Denghui 21818696@zju.edu.cn
Stomatology Hospital, School of Stomatology, Zhejiang University School of Medicine, Zhejiang Provincial Clinical Research Center for Oral Diseases, Key Laboratory of Oral Biomedical Research of Zhejiang Province, Cancer Center of Zhejiang University, Han , Hangzhou, Zhejiang , China
Castro Alves Ricardo
Electronic publication date: 2025 Jan 30
Publication date: 2025
Volume: 13
Electronic Location ID: e18678
Received 2024 Aug 30; Accepted 2024 Nov 19
Copyright: © 2025 Zheng et al.
Copyright year: 2025
Copyright holder: Zheng et al.
License: This is an open access article distributed under the terms of the Creative Commons Attribution License, which permits unrestricted use, distribution, reproduction and adaptation in any medium and for any purpose provided that it is properly attributed. For attribution, the original author(s), title, publication source (PeerJ) and either DOI or URL of the article must be cited.
License URL: https://creativecommons.org/licenses/by/4.0/

Keywords: Resident, 24-hour on-call, Pulpitis

Funding: Zhejiang University Comprehensive Deepening Reform Project in 2024 sgba2404 This work was supported by Zhejiang University Comprehensive Deepening Reform Project in 2024 with Project Grant No: sgba2404. The funders had no role in study design, data collection and analysis, decision to publish, or preparation of the manuscript.

==============================
Objectives

To analyze the 24-hour on-call experience and factors influencing the treatment decisions of a dental resident for dental emergencies, particularly pulpitis, during on-call hours.

Methods

This retrospective study was conducted at a public stomatology hospital from January 1 to December 30, 2023. Each consultation was documented, recording the date and time, patient age and gender, diagnosis, and any emergent interventions. Statistical analyses were conducted using univariate analysis to explore the association between various factors and the incidence of dental interventions for pulpitis, with significance set at p < 0.05.

Results

Over 1 year, 81 residents from seven specialties managed 2,717 consultations during 365 instances of 24-h call duty. The busiest months were October (n = 297). Most consultations occurred during extended hours (1,856 consultations) compared to normal hours (8:00–17:00) (861 consultations). The busiest consultation periods were between 20:00 and 22:00. Pulpitis was the most frequently diagnosed condition (n = 988). Univariate analysis showed no significant impact of patient gender (p = 0.896) or age (p = 0.632) on the likelihood of receiving a dental intervention. However, consultations during extended hours were twice as likely (OR = 2.028, 95% CI [1.510–2.723]) to result in no intervention compared to normal hours. Endodontics and pediatric dentistry residents were more likely to perform interventions compared to other specialties, with postgraduate year (PGY) six residents being less likely to perform interventions compared to PGY4 residents.

Conclusion

Residents exhibit lower willingness to perform dental interventions during extended working hours and in higher grade levels, with significant variability across different specialties. Enhanced training and fatigue risk management for residents may help to ensure effective patient care during on-call hours.

Introduction

Emergency dentistry plays a crucial role in the current healthcare system (Mahajan et al., 2019). With the rapid advancement of dental medicine, emergency dentistry has gradually evolved into a comprehensive discipline that addresses the acute manifestations of various oral diseases, encompassing endodontics, oral and maxillofacial surgery, prosthodontics, pediatric dentistry, orthodontics, and implantology (Sommacal et al., 2023). It provides urgent diagnostic and therapeutic services for patients experiencing sudden or worsening oral health conditions, focusing primarily on acute pain management, preventing disease progression, and ensuring the quality of life for patients (Hell et al., 2022).

Patients in dental emergencies often require urgent treatment due to acute facial or dental pain, bleeding, trauma, or other conditions (Franciscatto et al., 2020). The 24-hour on-call system ensures that timely medical services are available to patients at any time. When an emergency arises, the on-call physician and medical team can respond swiftly, providing immediate rescue and treatment (Alsharawneh & Maddigan, 2021). In the face of pain, patients often experience panic, anxiety, and other negative emotions (Kong, 2024). On-call medical staff accompany patients around the clock, offering care and support on a psychological level.

For residents, being on call in the dental emergency department is considered particularly stressful (Pietrement et al., 2023). Despite potential fatigue and stress, on-call shifts enhance residents’ knowledge, skills, and coping abilities. They improve independent thinking, decision-making, teamwork, communication, and motivation to learn, enabling them to provide higher quality and more efficient medical services (Navarro-Pérez et al., 2020). This continuous professional development enables them to provide higher quality and more efficient medical services to patients (Burnett et al., 2023). Arranging work hours reasonably, providing necessary support, and focusing on physicians’ well-being during on-call periods help residents handle emergencies efficiently. This includes addressing acute dental pain, trauma, and other urgent conditions, as well as improving communication with anxious patients, thus fostering professional growth (Cygler, Page & Ginsburg, 2021).

The 24-hour on-call system faces challenges such as insufficient medical skills, high workload, mental stress, and emotional fatigue, which can decrease residents’ enthusiasm and lead to thoughts of evading duties (Alsohime, 2019). Consequently, patients may not receive effective treatment or may suffer serious consequences due to delayed care (Kolnes et al., 2023).

The primary objective of this study is to quantitatively assess the workload, willingness to work, and the influence of specialty and working hours on the clinical duties of fourth- to sixth-year dental residents during on-call shifts in a dental emergency department. We hypothesize that the specialty of the resident and the duration of on-call hours have no significant impact on the willingness to perform dental intervention. Through this analysis, we aim to provide evidence-based insights that can inform future policies on resident workload management. Ultimately, it is expected that the findings will contribute to optimizing the current on-call system, improving resident satisfaction, and enhancing patient care in dental emergency settings.

Materials and Methods

Database creation and outcome

The project was approved by the local Ethics Committee (Stomatology Hospital, School of Stomatology, Zhejiang University School of Medicine, 2023-127(R)). The research has received a waiver of informed consent from the local Ethics Committee. The research was conducted from January 1, 2023, to December 30, 2023. During this period, the primary author recorded a “call log” for each on-call instance. The call log included the following data points for each consultation: (1) date and time of consultation; (2) patients’ age and gender; (3) diagnosis; and (4) any emergent or urgent interventions within 24 h of consultation or admission. Interventions for pulpitis included procedures such as pulp exposure, pulpotomy, root canal preparation, or single-visit root canal treatment. The collected data were entered into a secure, password-protected database. Double-entry verification was used to minimize data entry errors, and any discrepancies were resolved by cross-checking with the original call logs. Data were de-identified to maintain patient confidentiality, with unique numerical codes assigned to each patient entry. The final dataset was stored in a secure, encrypted file, accessible only to the research team.

Resident on-call system

The present study took place within a dental residency program at the emergency department of a large public stomatology hospital. The duration of on-call duties for residents varies by training level: PGY4–PGY6 residents handle primary call, while secondary and tertiary backup calls are managed by attending and chief physicians, respectively. At our institution, 81 residents from seven specialties (endodontics, periodontics, prosthodontics, oral and maxillofacial surgery, pediatric dentistry, implant dentistry, and orthodontics) share primary call responsibilities. Each primary call resident covers all consultations for a 24-h period (08:00 to 08:00 the next day). This includes managing all consultations, admissions, and addressing concerns for existing in-house patients during this time. “Consultation” refers to any patient seen by the dental resident upon request from another service. Day-call duties are equally distributed among junior residents rotating at the facility. During working hours (08:00 to 17:00), other departments are also open, including holidays and weekends, while only the emergency department operates during extended hours (17:00 to 08:00 the next day).

Statistical analysis

Continuous variables, such as patient age, were summarized using median and range, while categorical variables, such as gender and type of intervention, were presented as proportions. Before statistical analysis, continuous data were checked for normality using the Shapiro-Wilk test. Any non-normally distributed continuous data were subsequently analyzed using non-parametric methods. Missing data points were handled using pairwise deletion, ensuring that only complete cases were used for univariate analysis. Both univariate and multivariate analyses were conducted to examine the relationships between covariates and the likelihood of performing dental interventions in patients diagnosed with pulpitis. The covariates analyzed included sex, age, normal vs extended working hours, specialty, and postgraduate year (PGY) level. Univariate analysis was initially performed to assess the association of each variable independently with the primary outcome of dental intervention. Odds ratios (OR) and 95% confidence intervals (CI) were reported for each covariate. Categorical variables were analyzed using the chi-squared test, while continuous variables were analyzed using independent t-tests or Mann-Whitney U tests, depending on the normality of the data. Subsequently, multivariate analysis was employed to control for potential confounding factors and to assess the combined effects of the covariates on the likelihood of performing a dental intervention. A logistic regression model was used for this purpose, with ORs and 95% CIs calculated to provide a more comprehensive understanding of the factors influencing dental interventions. All statistical analyses were conducted using Microsoft Excel and SPSS version 22 (IBM Corp, Armonk, NY, USA), with a p-value of less than 0.05 considered statistically significant.

Results

Residents, call volume and consultations characteristics

Figure 1 illustrates the residents’ characteristics (Fig. 1A), 24-h call volume (Fig. 1B), and the consultations handled by primary authors for each specialty and PGY level (Fig. 1C). Over the course of 1 year, 81 residents from seven different specialties were responsible for 365 instances of 24-h call duty, managing a total of 2,717 consultations. The average number of consultations per 24-h call was 7.44. Among the specialties, residents in oral and maxillofacial surgery and endodontics had the highest numbers and were responsible for more 24-h call duties and consultations compared to other specialties.

Figure 1 Residents’ call volume and consultation characteristics categorized by specialty and post-graduate year (PGY) level.

(A) Distribution of resident characteristics by specialty and PGY level. (B) 24-h call volume handled by the primary author, categorized by specialty and PGY level. (C) Number of consultations managed by the primary author, categorized by specialty and PGY level.

Date, and time of consultation

On average, 226 new consultations (ranging from 142 to 297) were recorded monthly during the study period. October had the highest number of consultations (n = 297), followed by September (n = 280), April (n = 257), June (n = 253), and March (n = 247; Fig. 2). The busiest season was fall, followed by spring, summer, and winter. Each month, new consultations during extended working hours surpassed those during normal working hours. Throughout the study, there were 1,856 consultations during extended hours, compared to 861 during normal hours.

Figure 2 Number consultations by month (orange and blue bars represent normal working hours and extended working hours; the black dashed line represents average number in any month (226)).

When analyzing the number of consultations per 24 h, the average number of consultations per hour was 113 ± 72. Over time, there was an overall upward trend in consultations (y = 7.9839x + 21.393) (Fig. 3). Interestingly, the hours from 20:00 to 21:00 (n = 262) and from 21:00 to 22:00 (n = 276) had the highest number of consultations within the 24-h period. The number of consultations between 19:00 and 23:00 was more than one standard deviation above the mean, while the number of consultations between 4:00 and 6:00 was less than one standard deviation below the mean (Fig. 3).

Figure 3 Number of consultations seen per hour throughout the study period.

The dark solid line represents the average number of consultations per hour (113). The black dashed line indicates the standard deviation from the mean (±72). The blue dashed line.

Diagnosis and procedures

Figure 4 shows the various diagnostic subcategories. The five most common diagnoses were pulpitis (n = 988), apical periodontitis (n = 452), acute pericoronitis (n = 411), dental trauma (n = 242), and oral and maxillofacial trauma (n = 191).

Figure 4 Diagnostic subcategories (note that the sum is >2,717 because some patients had more than 1 diagnosis).

Since pulpitis was the most common diagnosis, univariate analysis (Table 1) was employed to investigate the association between the documented factors and the incidence of dental intervention. The patient’s gender (p = 0.805) and age (p = 0.775) did not have a significant impact on whether a dental intervention was performed. However, the likelihood of not performing a dental intervention for patients seen during extended working hours is 1.814 times (95% CI [1.389–2.370]) that of normal working hours.

Table 1 Univariate analysis of dental interventions performed in pulpitis patients.

Covariate	Yes	No	OR (95% CI)	p-value	
Sex					
Male	293 (49.6%)	200 (50.4%)	–	–	
Female	298 (50.4%)	197 (49.6%)	0.968 [0.751–1.249]	0.805	
Age, years	31.21 ± 16.64	31.52 ± 16.82	1.001 [0.994–1.009]	0.775	
Normal or extended time					
Normal working time	267 (45.2%)	124 (31.2%)	–	–	
Extended working time	324 (54.8%)	273 (68.8%)	1.814 [1.389–2.370]	<0.001	
Major					
Endodontics	186 (31.5%)	53 (13.4%)	–	–	
Periodontics	42 (7.1%)	39 (9.8%)	3.259 [1.914–5.548]	<0.001	
Prosthodontics	61 (10.3%)	50 (12.6%)	2.877 [1.775–4.661]	<0.001	
Oral and maxillofacial surgery	72 (12.2%)	104 (26.2%)	5.069 [3.303–7.779]	<0.001	
Pediatric dentistry	122 (20.6%)	24 (6.0%)	0.690 [0.405–1.177]	0.174	
Implant dentistry	72 (12.2%)	66 (16.6%)	3.217 [2.046–5.057]	<0.001	
Orthodontics	36 (6.1%)	61 (15.4%)	5.947 [3.561–9.929]	<0.001	
Level					
PG4	215 (36.4%)	110 (27.7%)	–	–	
PG5	238 (40.3%)	139 (35.0%)	1.142 [0.837–1.558]	0.404	
PG6	138 (23.4%)	148 (37.3%)	2.096 [1.512–2.905]	<0.001	

Except for pediatric dentistry residents (OR = 0.690, 95% CI [0.405–1.177]), residents in other specialties were more likely to not perform dental interventions compared to Endodontics residents. Interestingly, PGY6 residents showed a higher likelihood of not performing dental interventions for patients with pulpitis compared to PGY4 residents (OR = 2.096, 95% CI [1.512–2.905]).

To offer deeper insight into the relationships between the factors studied, multivariate analysis (Table 2) was employed. Patients seen during extended working hours were 2.028 times more likely (95% CI [1.510–2.723], p < 0.001) to not undergo a dental intervention compared to those seen during normal working hours. Furthermore, the specialty of the resident also influenced the likelihood of performing dental interventions. For example, residents in oral and maxillofacial surgery had a significantly higher likelihood of not performing dental interventions (OR = 6.322, 95% CI [4.012–9.961], p < 0.001), while those in endodontics served as the reference group. Other specialties, such as orthodontics (OR = 5.549, 95% CI [3.285–9.374], p < 0.001) and Prosthodontics (OR = 3.527, 95% CI [2.137–5.822], p < 0.001), also had elevated odds of not performing dental interventions compared to endodontics. Additionally, PG6 residents were 2.378 times (95% CI [1.627–3.476], p < 0.001) more likely to not perform dental interventions compared to PG4 residents, suggesting that experience level plays a role in treatment decisions during on-call hours.

Table 2 Multivariate analysis of dental interventions performed in pulpitis patients.

Covariate	Yes	No	OR (95% CI)	p-value	
Sex					
Male	293 (49.6%)	200 (50.4%)	–	–	
Female	298 (50.4%)	197 (49.6%)	0.986 [0.742–1.298]	0.896	
Age, years	31.21 ± 16.64	31.52 ± 16.82	1.002 [0.994–1.011]	0.632	
Normal or extended time					
Normal working time	267 (45.2%)	124 (31.2%)	–	–	
Extended working time	324 (54.8%)	273 (68.8%)	2.028 [1.510–2.723]	<0.001	
Major					
Endodontics	186 (31.5%)	53 (13.4%)	–	–	
Periodontics	42 (7.1%)	39 (9.8%)	3.186 [1.825–5.564]	<0.001	
Prosthodontics	61 (10.3%)	50 (12.6%)	3.527 [2.137–5.822]	<0.001	
Oral and maxillofacial surgery	72 (12.2%)	104 (26.2%)	6.322 [4.012–9.961]	<0.001	
Pediatric dentistry	122 (20.6%)	24 (6.0%)	0.695 [0.401–1.202]	0.193	
Implant dentistry	72 (12.2%)	66 (16.6%)	2.669 [1.672–4.261]	<0.001	
Orthodontics	36 (6.1%)	61 (15.4%)	5.549 [3.285–9.374]	<0.001	
Level					
PG4	215 (36.4%)	110 (27.7%)	–	–	
PG5	238 (40.3%)	139 (35.0%)	1.266 [0.889–1.804]	0.192	
PG6	138 (23.4%)	148 (37.3%)	2.378 [1.627–3.476]	<0.001	

Discussion

In this study, the majority of on-call requests for dental residents were due to odontogenic infections. Numerous studies have shown that using emergency departments to treat odontogenic infections often results in uncertain care, failing to fundamentally resolve dental issues, and ultimately necessitating another visit to a dental clinic (Cohen et al., 2011). One study estimated that each admission for dental pain costs the healthcare system approximately $3,000 (Cohen et al., 2008). The high social and human costs associated with the 24-hour on-call system do not necessarily yield the anticipated treatment outcomes. Therefore, hospitals and medical institutions should prioritize the construction and management of the on-call system to ensure that on-call doctors and medical teams can fully perform their roles and provide higher quality and more efficient medical services to patients (Wu et al., 2021).

Previous studies have found that the volume of emergency patients is highest in July, as medical and dental interns begin their residency training programs, leading to a surge in patient outcomes and consultation demands with more novice providers starting to offer services (Young et al., 2011). This does not align with our findings. As shown in Fig. 2, the number of patient visits varies by month. The highest number of patient visits occurred in September and October, possibly due to the pleasant weather in eastern China during these months (Wang et al., 2016), leading to increased outdoor activities, more traumas, and a higher willingness to seek medical (Matsa et al., 2018). Additionally, the incidence of many diseases is seasonally related (Schmid et al., 2023). The primary diagnosis in our study was pulpitis, which some articles suggest is seasonally related (Zeng, Sheller & Milgrom, 1994), potentially explaining the higher number of emergency visits in September and October. February had the fewest emergency visits, which may be related to the traditional Chinese belief of not seeking medical treatment during the first month of the lunar year (Tan et al., 2020).

The highest number of patient visits during the day was concentrated at 21:00 and 22:00, with the fewest visits at 06:00. Overall, from 00:00 to 24:00, there was a gradual increase in patient visits (Fig. 3). Due to nighttime sympathetic excitation and parasympathetic inhibition, the symptoms of spontaneous pain from acute pulpitis are particularly pronounced at night (Liu & Shi, 2017; Parirokh et al., 2010; Zhang et al., 2020). Thus, the number of patients seeking treatment for pain significantly increases between 20:00 and 22:00. Additionally, patients with acute pericoronitis often choose to visit after dinner, around 20:00, to avoid missing work during the day when their pain and swelling are not too severe. Similar to many other studies, our research found that 06:00 had the fewest patient visits, likely because the normal outpatient clinics open at 08:00, and patients prefer to wait until then for treatment (Gordon et al., 2019). The number of patients was higher outside regular clinic hours (17:00–23:00) than during regular hours (08:00–17:00), likely because patients have various other departments to choose from during regular hours, such as endodontics and comprehensive dental care (Kang & Park, 2015; Pianucci & Longacre, 2022). Outside regular working hours, only the dental emergency department is open, leading to a concentration of patients.

The treated conditions included acute pulpitis, acute pericoronitis, acute apical periodontitis, among others, with acute pulpitis being the most common. In 2023, there were 988 cases of acute pulpitis in the emergency department, far exceeding the second most common condition, acute apical periodontitis, with 452 cases (Fig. 4). This trend is consistent with previous studies (Gemmell, Stone & Edwards, 2020; Spencer, 2017). After the removal of stimulating factors, pain usually subsides, and patients generally choose to visit regular outpatient clinics during normal working hours. However, those visiting the emergency department often have significant spontaneous pain or nighttime pain, indicative of acute, irreversible pulpitis (Cunha et al., 2022). Acute pulpitis is characterized by severe pain, spontaneous pain, and nighttime pain (Spencer, 2017). Treatment for acute pulpitis typically requires endodontic procedures such as pulpectomy and removal of the infected pulp to relieve pain (European Society of Endodontology, 2006; AAE Position Statement, 2021). However, during the data collection process, it was found that residents often did not perform treatment procedures for many patients with acute pulpitis. Therefore, to further analyze the relationship between the workload, willingness to work, specialty, and working hours of residents in the dental emergency department, we conducted a univariate analysis of the cases of acute pulpitis, examining whether residents performed treatment procedures for these patients.

As shown in Table 1, there was no significant difference in the occurrence of pulpotomy procedures among residents based on the gender or age of the patients. There was no observed reluctance to perform procedures on pediatric patients, likely due to the rotation of PG1-3 residents in pediatric dentistry contributing to this phenomenon (Rembeck et al., 2024; Tabatabaei et al., 2022). However, during extended working hours, the proportion of residents choosing not to perform pulpotomy procedures (68.8%) was significantly higher than during regular working hours (31.2%). We speculate that this may be due to residents’ physical and mental states, and out of consideration for medical safety, they may prefer prescribing medication to avoid the risks associated with invasive procedures like pulpotomy. Doctors need to follow their risk profile when choosing a treatment for their patients (Buturovic, 2023; Eeckhoudt, Lebrun & Sailly, 1985).

Additionally, we found differences in the willingness to perform procedures among residents from different specialties and seniority levels. Residents specializing in endodontics and pediatric dentistry had a relatively higher willingness to treat, while those from other specialties had a lower willingness. This is related to the scope of their specialty (Alshoraim et al., 2018; Guerrero et al., 2017). Endodontics and pediatric dentistry residents, working in their respective departments during PG 4-6, are familiar with and proficient in the diagnosis, treatment, and pulpotomy procedures for acute pulpitis due to their extensive daily experience. Consequently, they have the skills and confidence to handle such cases (Sun et al., 2024). Conversely, residents in oral and maxillofacial surgery, implantology, orthodontics, and prosthodontics, despite rotating through endodontics and pediatric dentistry during PG 1-3, have limited exposure to these patients in their senior years. Thus, they may exhibit conservatism and lack of confidence in performing pulpotomy procedures for acute pulpitis (Nguyen et al., 2023; Soh, Lim & Yip, 2020). Moreover, senior residents tend to be more cautious in medical procedures (Peters et al., 2006). As shown in Table 1, the higher the seniority, the higher the proportion of residents unwilling to perform pulpotomy procedures. At the same time, they possess better communication skills, which allows them to effectively comfort and persuade patients with less strong treatment desires to seek care during regular hours or at other specialty clinics (Gemmell, Stone & Edwards, 2020).

The analysis above indicates that residents exhibit lower work willingness during extended working hours and in higher grade levels, with variations in willingness across different specialties. To ensure patients receive effective treatment, it is essential to strengthen the training of on-call residents. Clinical guidance from senior residents plays a crucial role in enhancing treatment outcomes while maintaining patient safety, as supported by Meade et al. (2022). Senior residents, with their greater experience and decision-making capabilities, are often responsible for overseeing junior residents and making critical decisions during emergency shifts (Chiu et al., 2017). However, our findings indicate that senior residents are less likely to perform direct interventions, particularly during extended working hours. This could be attributed to their supervisory roles and a tendency to delay interventions until regular working hours when more resources and support staff are available. To address this issue, it is essential to provide additional training that focuses on balancing their supervisory responsibilities with the need for timely interventions, especially in emergency situations (Huda, Faden & Goldszmidt, 2017). Training could emphasize decision-making under pressure, with particular attention to emergency cases that arise during extended working hours. Ensuring that senior residents feel supported to perform interventions when necessary could improve patient outcomes and increase their willingness to act in these critical situations (Hudkins, Helmer & Smith, 2009). Furthermore, implementing fatigue risk management programs for residents is equally crucial. Fatigue has a significant impact on residents’ physical, cognitive, and emotional states, potentially reducing their efficiency and increasing the risk of medical errors. Fatigue management programs can include strategies such as optimized shift scheduling, mandatory rest periods, and regular assessments of fatigue levels to ensure that residents are functioning at their best. By addressing fatigue proactively, these programs can support both resident well-being and patient safety (Kassam, Cowan & Topps, 2019). The extended 24-hour on-call periods, with the highest patient volume occurring after 19:00, contribute to significant work stress. Studies have shown that the work stress for on-call residents often exceeds that of emergency department physicians, influenced by factors such as patient volume, chronic disease care, and high workload perception (Tür et al., 2016). Resident physicians’ mental and physical health must be addressed (Volpp et al., 2013). Future studies should explore the sources of psychological stress, analyzing the causes of resident burnout considering residents’ conditions, work hours, and patient load (Schwartz et al., 2021). Reducing shift lengths when patient numbers exceed a certain threshold might be necessary (Nomura et al., 2016). Shorter resident duty hours could improve resident-based outcomes (Sephien et al., 2023). Additionally, the effects of extended duty hours on ambulatory blood pressures in internal medicine, pediatric, and med-peds residents (Czeisler et al., 2023). Furthermore, considering that most dental emergency cases are related to endodontics, ensuring appropriate training for all residents in this specialty is vital. Alternatively, limiting emergency shifts to residents from endodontics and pediatric dentistry may be beneficial.

This study did not include a questionnaire survey, preventing us from understanding residents’ subjective reasons and difficulties in choosing whether to perform treatments on dental emergency patients, as well as their burnout reasons during emergency shifts. Additionally, the study only analyzed patient data from January to December 2023, making the study period relatively short and possibly introducing errors. For example, some patients may choose not to proceed with treatment after being registered, which could lead to an underestimation of the proportion of untreated cases in the analysis. One of the key limitations of this study is its retrospective design, which inherently limits our ability to control for certain confounding factors, such as residents’ fatigue and workload. Since the data were collected retrospectively, there was no direct measurement of fatigue levels or workload intensity during the shifts. As a result, we were unable to include more precise metrics for these factors, which could have influenced residents’ decision-making during on-call shifts. Instead, general indicators such as working hours and the number of consultations were used to infer workload and fatigue, but these may not fully capture the complexities of the situation. Additionally, while our study focused on interventions related to pulpitis due to its prevalence and the standardization of its treatment, the analysis did not cover other dental interventions, which may present more variability in treatment decisions. A prospective study design in the future could address these issues by incorporating real-time fatigue assessments and a broader analysis of different types of dental interventions.

Conclusions

Dental residents exhibit lower willingness to perform dental interventions during extended working hours and in higher grade levels, with significant variability across different specialties. Enhanced training and fatigue risk management for residents may help to ensure effective patient care during on-call hours.

Supplemental Information

Supplemental Information 1 Consults diagnosed with pulpitis.

Each data point indicates a consult with the date and time, patient age and gender, diagnosis, and any emergent interventions.

Supplemental Information 2 Codebook.

Additional Information and Declarations

Competing Interests

The authors declare that they have no competing interests.

Author Contributions

Jiaoer Zheng conceived and designed the experiments, performed the experiments, analyzed the data, prepared figures and/or tables, authored or reviewed drafts of the article, and approved the final draft.

Ji Xu performed the experiments, analyzed the data, prepared figures and/or tables, authored or reviewed drafts of the article, and approved the final draft.

Denghui Zhang analyzed the data, authored or reviewed drafts of the article, and approved the final draft.

Human Ethics

The following information was supplied relating to ethical approvals (i.e., approving body and any reference numbers):

Stomatology Hospital, School of Stomatology, Zhejiang University School of Medicine granted Ethical approval to carry out the study within its facilities (Ethical Application Ref: 2023-127(R)).

Data Availability

The following information was supplied regarding data availability:

The raw measurements are available in the Supplemental File.

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
