# Peer review of "An analysis of the 24-hour on-call experience and treatment decision of a dental resident, a retrospective study"

_PeerJ, doi:10.7717/peerj.18678_

## Round 0.1 · original submission · Major Revisions

Please re-submit the manuscript paying particular attention to the questions raised by reviewer 1

Reviewer 1 ·

Basic reporting

• The manuscript is generally well-written and uses professional language, although some sentences are complex and may benefit from revision for clarity.
• The introduction and background provide good context.
• The figures and tables are relevant, but the labeling and descriptions could be more detailed, particularly in Figure 1 and Figure 3. The table of univariate analysis (Table 1) is clear but could benefit from more explanation in the text.
• The raw data is supplied, but more descriptive information about how the data was handled would be useful for replication purposes.

Experimental design

• The study fits within the scope of the journal. However, the research question and objectives should be more explicitly stated.
• The methodology is sufficient for a retrospective analysis, but the study design lacks details on how confounding factors were controlled. More information is needed about how residents' fatigue and workload were measured and factored into the analysis. The analysis of other dental interventions is also missed.

Validity of the findings

• The findings are interesting and relevant to improving dental emergency care but could be further strengthened by deeper analysis.
• The discussion around the residents' reluctance to perform interventions during non-working hours would benefit from comparing the results to other studies that have explored similar issues.
• The statistical analysis is sound, but a multivariate analysis would offer deeper insight into the relationships between the factors studied.

Additional comments

1. The language need to be polished. For example, “The peak consult times” and “The most common diagnoses were pulpitis ” in the abstract. Please check the writings carefully.
2. How did the authors account for potential confounding variables, such as resident fatigue or workload, when analyzing intervention rates?
3. The authors suggest that senior residents were less likely to perform interventions. Could you provide more context for this finding, and what steps could be taken to address this issue?
4. Can the authors elaborate on how the time of day influenced residents' willingness to perform interventions, and whether these findings are consistent with other studies in dental emergency care?
5. Why thers is only univariate analysis of dental interventions performed in pulpitis patients, because the rate and emergency level of dental trauma or pericoronitis is also high.
6. The study provides valuable insights into resident decision-making in emergency dental care, highlighting the challenges residents face during on-call shifts. However, the manuscript would benefit from a clearer focus on how these findings could impact clinical practice or resident training.

Reviewer 2 ·

Basic reporting

- English could be reviewed
- Consider replacing the word “consult” by “consultation” or “appointment”
- Line 200 - …. Non-working hours … Consider the other designation that you use in the text: extended or extending hours that is an expression more understandable. In fact, they are working… Check in all the article (ex. Line 225, 254)

Table 1

1. Consider replacing the word “consult” by “consultation” or “appointment”
2. Attention to Table 1: Not all rows are visible. The same happens in the Word document as in the pdf. Ex. 0.986(0.742- We don’t see all the interval.
3. Title of Table 1: Remove capital letters on Dental Interventions Performed
4. Table 1: Consider Age (years) and place it at the same left level as Sex or Normal or extending hours
5. The introduction of discrete lines would help the division of the information

Experimental design

No comment

Validity of the findings

No comment

Additional comments

Overall interesting paper showing a real context of on-call appointments in an emergency scenario.
Needing some review.

Abstract

1. Consider replacing the word “consult” by “consultation” or “appointment”
2. Line 25 – Normal hours – Clarify indicating the range

Remaining text

1. Consider replacing the word “consult” by “consultation” or “appointment” in all the text or alternate between the above two.
2. Line 125 – Diagnosis and Procedures
3. Line 142 - …. on-call pages .. Clarify the meaning of “pages” or improve the designation
4. Line 200 - …. Non-working hours … Consider the other designation that you use in the text: extended or extending hours that is an expression more understandable. In fact, they are working… Check in all the article (ex. Line 225, 254)

Table and figures

1. Consider replacing the word “consult” by “consultation” or “appointment”
2. Attention to Table 1: Not all rows are visible. The same happens in the Word document as in the pdf. Ex. 0.986(0.742- We don’t see all the interval.
3. Title of Table 1: remove capital letters on Dental Interventions Performed
4. Table 1: Consider Age (years) and place it at the same left level as Sex or Normal or extending hours
5. The introduction of discrete lines would help the division of the information

---

## Round 0.2 · accepted · Accept

The authors have addressed all of the reviewer's comments. The manuscript is now ready for publication

Reviewer 1 ·

Basic reporting

Clarity and Language: The revised manuscript is clear, well-structured, and written in professional English. Changes to terms like "consult" to "consultation" and "non-working hours" to "extended working hours" improve readability and consistency.
Introduction and Context: The introduction provides a solid background, outlining the relevance of emergency dental care and the challenges faced by residents. The study's objectives are now explicitly stated and align with the presented data.
Figures and Tables: The revisions to figure labels and descriptions enhance clarity. The detailed updates to Table 1 improve its readability and organization.
Data Sharing: The detailed descriptions of data handling and analysis methods ensure replicability. The raw data is supplied and adequately described.

Experimental design

Scope and Objectives: The study fits well within the scope of PeerJ. The research question—focusing on workload, willingness to work, and specialty-related variability during on-call shifts—is relevant and meaningful.
Methodology: The expanded methods section clearly describes the retrospective study design, database creation, and statistical analysis, including the addition of multivariate analysis.
Control of Confounders: While the retrospective nature limits full confounding control, the authors have adequately addressed this limitation in the discussion and included multivariate analyses to strengthen their findings.
Ethical Considerations: The manuscript confirms approval by an ethics committee and a waiver of informed consent, meeting ethical research standards.

Validity of the findings

Data Interpretation: The findings are well-supported by statistical analysis and align with the study objectives. The addition of multivariate analysis strengthens the conclusions about factors influencing intervention likelihood.
Novelty and Relevance: The results provide valuable insights into dental emergency care and resident decision-making, with implications for training and policy development.
Limitations: The discussion appropriately acknowledges limitations, including the short study period and reliance on retrospective data, which were well-addressed in the revised manuscript.
Conclusions: The conclusions are logical, supported by the data, and highlight actionable recommendations, such as improving resident training and fatigue management.

Additional comments

The revisions effectively address prior concerns, particularly in clarifying the research question, enhancing statistical analysis, and improving figure and table descriptions.
The inclusion of multivariate analysis offers a more comprehensive understanding of the factors affecting resident decision-making during on-call shifts.
The discussion now provides a broader context by comparing findings to existing literature, further strengthening the manuscript.

Reviewer 2 ·

Basic reporting

Suggestions were followed

Experimental design

Suggestions were followed

Validity of the findings

No comments